# A Comparison of the Neuromodulation Effects of Frontal and Parietal Transcranial Direct Current Stimulation on Disorders of Consciousness

**DOI:** 10.3390/brainsci13091295

**Published:** 2023-09-08

**Authors:** Xiaoping Wan, Yong Wang, Ye Zhang, Weiqun Song

**Affiliations:** 1Department of Rehabilitation Medicine, Xuan Wu Hospital, Capital Medical University, No. 45 Chang Chun Street, Beijing 100053, China; wxp13366405536@163.com (X.W.); 18330180615@163.com (Y.Z.); 2Zhuhai UM Science & Technology Research Institute, No. 1889 Huandao East Road, Zhuhai 519031, China; wy599580@163.com

**Keywords:** disorders of consciousness (DoC), tDCS, G_PCMI, EEG

## Abstract

Frontal transcranial direct current stimulation (tDCS) and parietal tDCS are effective for treating disorders of consciousness (DoC); however, the relative efficacies of these techniques have yet to be determined. This paper compares the neuromodulation effects of frontal and parietal tDCS on DoC. Twenty patients with DoC were recruited and randomly assigned to two groups. One group received single-session frontal tDCS and single-session sham tDCS. The other group received single-session parietal tDCS and single-session sham tDCS. Before and after every tDCS session, we recorded coma recovery scale-revised (CRS-R) values and an electroencephalogram. CRS-R was also used to evaluate the state of consciousness at 9–12-month follow-up. Both single-session frontal and parietal tDCS caused significant changes in the genuine permutation cross-mutual information (G_PCMI) of local frontal and across brain regions (*p* < 0.05). Furthermore, the changes in G_PCMI values were significantly correlated to the CRS-R scores at 9–12-month follow-up after frontal and parietal tDCS (*p* < 0.05). The changes in G_PCMI and CRS-R scores were also correlated (*p* < 0.05). Both frontal tDCS and parietal tDCS exert neuromodulatory effects in DoC and induce significant changes in electrophysiology. G_PCMI can be used to evaluate the neuromodulation effects of tDCS.

## 1. Introduction

With the advancement of emergency techniques and intensive care, the survival rate of patients with severe brain injuries has improved. Before returning to family and society, these patients will experience a process of autonomous eye opening, regaining consciousness, and functional rehabilitation. Patients who do not fully regain consciousness are known to have disorders of consciousness (DoC), for which there are two main categories: unresponsive wakefulness syndrome (UWS) and minimally conscious state (MCS). The incidence rate of DoC varies geographically. A previous study estimated that there were 10,000–25,000 adults with UWS and 280,000 adults with MCS in the United States [1]. In some European countries and Japan, the prevalence of UWS varied from 0.2 to 3.4 per 100,000, while that of MCS was 1.5 per 100,000 [2]. Patients with a lack of consciousness may require a longer course of recovery in terms of functional rehabilitation and have more complications than those with consciousness [3]. Thus, patients with a lack of consciousness are associated with increased levels of difficulty with regard to nursing care, family burden, and medical input [4]. Thus, identifying new methods to accelerate the process of regaining consciousness in DoC is vital.

To date, many interventions have been applied to treat DoC [5]. Transcranial direct current stimulation (tDCS) is a promising intervention for DoC; this method has several advantages, including the fact that it is non-invasive, economical, and has few ethical issues. tDCS mainly modulates spontaneous brain activity and cortical excitability, but does not directly induce neuronal action potentials. tDCS achieves these effects by delivering a weak current through two electrodes to regulate the subthreshold neuronal membrane potential. The anodic electrode exerts excitatory effects, while the cathodic exerts inhibitory effects [6]. tDCS was first reported for the treatment of DoC in 2014 [7,8]. Since then, an increasing number of studies have been carried out to verify its effects in patients with DoC [9,10]. Identifying a key target is one of the most important parameters to consider when designing a stimulation protocol. However, limited by small sample sizes and heterogeneous outcome measures, it is difficult to draw a conclusion from the existing literature [11]. The dorsolateral prefrontal cortex (DLPFC) [12,13,14,15,16,17,18,19,20] and primary motor cortex (M1) [21] were the original target choice for tDCS. Analysis demonstrated that patients with DoC could benefit from tDCS, especially those who underwent left DLPFC stimulation [7,8,16,19,20,22,23,24,25]. Recently, researchers tried to apply tDCS in the precuneus (a part of the parietal cortex); these studies reported a range of positive effects [26,27,28,29]. A number of studies have been published in relation to the application of frontal tDCS and parietal tDCS in the treatment of DoC; however, no studies have compared the difference between these two methods in a single protocol.

When it comes to the assessment of levels of consciousness and the effects of tDCS, behavioral scales are the most commonly used assessment instruments. However, behavioral assessment have the disadvantages of high subjectivity and a high misdiagnosis rate [30,31,32]. Electrophysiological evaluation is an objective and precise method for evaluating the level of consciousness in patients with DoC. Compared with neuroimaging methods, electrophysiology has the advantages of being cost-effective, readily available, and exhibits an excellent resolution that can detect changes in brain activities at the millisecond scale. Thus, the majority of studies applied a combination of behavioral scales and electrophysiology to evaluate the effects of tDCS in patients with DoC.

Recent studies have demonstrated that stimulating the frontal or parietal cortex can help patients with DoC to improve their level of consciousness. However, few studies have compared the difference between frontal tDCS and parietal tDCS in terms of their relative modulatory effects on DoC. In this study, we applied frontal and parietal tDCS on patients with DoC and used an electroencephalogram (EEG) combined with coma recovery scale-revised (CRS-R) to evaluate outcomes and effects. The aim of this study was to compare the neuromodulation effects of frontal and parietal tDCS and provide a basis for determining the mechanism underlying these two targets.

## 2. Materials and Methods

### 2.1. Subjects

Twenty patients with DoC (twelve MCS and eight UWS patients) were recruited from the Department of Rehabilitation Medicine at Xuan Wu Hospital, Beijing, China. The inclusion criteria of the study were as follows: (1) patients who were diagnosed with UWS or MCS by repeated CRS-R, (2) patients who were in a stable condition, (3) patients with a duration of more than 1 month, and (4) patients who showed no signs of improvement in consciousness for more than 1 month. The exclusion criteria were as follows: (1) patients with a history of neurological diseases or psychiatric disorders, (2) patients taking drugs or undergoing any other treatments that may have affected cortical excitability, (3) patients with epilepsy or frequent uncontrolled spontaneous movements, and (4) patients with pacemakers, arterial clips, and other metal implants in the body. Patients with MCS and UWS were randomly divided into two groups. Each group contained ten patients (six MCS and four UWS). One group received prefrontal tDCS, while the other received parietal tDCS. Key demographics (etiology, age, gender, and course of disease) of the two groups are given in Table 1.

### 2.2. Experimental Design

All patients with DoC received one anodal tDCS and one sham stimulation. Two sessions of stimulation were separated by a 24 h wash-out time; this was long enough for potential tDCS effects to disappear. Before and after each session of real tDCS or sham stimulation, patients received CRS-R and EEG assessments to evaluate the effects of treatment. Then, the patients underwent a 9–12-month follow-up with CRS-R evaluation (Figure 1).

### 2.3. Protocol for tDCS Stimulation

Stimulation was delivered with an Eldith DC-stimulator (neuroConn GmbH, Ilmenau, Germany) and involved the generation of one current and two electrodes (one anode and one cathode). The electrodes were two round pasters, each with a radius of 3 cm. In the tDCS protocol, the anode was placed over F3 in the frontal group and Pz in the parietal group. The cathode of the two groups was placed at Fp2. The F3, Pz, and Fp2 regions were located with an International 10/20 EEG System. Stimulation to the tDCS groups was delivered over 20 min with 2 mA current and 15 s ramp-up and ramp-down periods. The parameters of the sham stimulation were the same and included tDCS, except that the intensity was maintained at 0 mA after 15 s of the ramp-up period.

### 2.4. Behavioral Assessment

CRS-R was applied to assess the state of consciousness in patients with DoC. CRS-R contains six subscales that evaluate the consciousness of patients with DoC on the basis of six aspects: auditory function, visual function, motor function, oromotor/verbal function, communication, and arousal. CRS-R can reflect signs related to the brainstem, subcortical activity, and cortical activity; the total score ranges from 0 to 23. Items on the lower part of each subscale are responsible for relatively low scores and represent reflex function, while items on the upper part of each subscale are responsible for relatively high scores and represent cognitive function. CRS-R is the most sensitive tool of all established behavioral scales and can help identify subtle differences between patients with UWS and MCS. Repetitive CRS-R assessment was applied to patients with DoC over three days. Each assessment was performed with patients in good condition and the highest scores were considered the basis for diagnosis. All CRS-R assessments were performed by one experienced physician.

### 2.5. EEG Measurement

Prior to data acquisition, all patients were required to wash their hair to remove scalp oil to reduce skin impedance. EEG data were recorded using a 64-lead amplifier (Brain Products, Gilching, Bavaria, Germany). A 64 Ag/Cl electrode cap was applied to patients with DoC, and EEG signals were displayed on a BP viewer in real time. Throughout the process, patients were required to be awake (keep their eyes open) and surroundings were kept quiet. To ensure data quality, the skin/electrode impedance of each recording electrode was maintained <5 KΩ. The experiment was suspended if swallowing, sweating, or uncontrolled head movement occurred.

### 2.6. EEG Pre-Processing and Processing

Pre-processing was carried out in EEGLAB, a plug-in unit of Matlab R2021b (Mathworks, Natick, MA, USA). First, we imported the original data into EEGLAB with a down-sampling rate of 250 Hz. Then, we configured the channel locations and set the filter as a band-pass mode of 1–45 Hz. Then, we checked the whole data and replaced the bad electrodes. Next, we segmented the data into epochs of 4 s and checked every epoch to delete the data of poor quality. Then, we ran independent component analysis to reject electromyographic (EMG) and electrooculographic (EOG) activity. Finally, we obtained the average reference data. The entire process is given in Figure 2.

### 2.7. Genuine Permutation Cross-Mutual Information

Values derived from permutation cross-mutual information (PCMI) can measure the functional connectivity of different cortices and estimate integrated information of the brain based on EEG data. Genuine permutation cross-mutual information (G_PCMI), originally proposed by Liang et al. [20], is an improved version of the mutual information algorithm and can correct spurious correlations by statistically testing data from the original PCMI and surrogate PCMI. The specific steps of the calculation were as follows (the structure is displayed in Figure 2):

Step 1: By applying a phase space reconstruction procedure, the vectors X_t_ [x_t_, x_t+τ_, …, x_t++mτ_] and Y_t_ [y_t_, y_t+τ_, …, y_t++mτ_] were constructed with embedding dimension m and lag τ. In this study, we defined m = 3 and τ = 1. The x_t_ and y_t_, t = 1, 2, …, n, refer to two given EEG data; t refers to the sample point.

Step 2: Symbol vectors (S_n_ and S_q_) were defined based on element sorting of the vectors, X_t_ and Y_t_, respectively. When this embedding dimension (m) was 3, then the sum of the symbolic vector type was 3, theoretically.

Step 3: p (x) and p (y) were defined as the marginal probability distribution functions of x_t_ and y_t_ with specific symbolic vector types, respectively. The entropy of x_t_ and y_t_ were as follows:(1)$$H\left(X\right)=-\sum_{j=1}^{J}p_{j}{logp}_{j}$$
(2)$$H\left(Y\right)=-\sum_{j=1}^{J}p_{j}{logp}_{j}$$

Step 4: based on the joint probability function, the joint of entropy was calculated as follows:(3)$$H\left(X,Y\right)=-\sum_{x\in X}\sum_{y\in Y}p\left(x,y\right)\log p(x,y)$$

Step 5: the PCMI of the time series x_t_ and y_t_ was calculated using the following equation.
(4)$$PCMI\left(X;Y\right)=H\left(X\right)+H\left(Y\right)-H(X,Y)$$

Step 6: The time series of x_t_ and y_t_ were used to generate the surrogate time series $x_{t}^{surr}$ and $y_{t}^{surr}$ through the iterative amplitude-adjusted Fourier transform method. In this study, 50 surrogate time series were used to calculate the corresponding PCMI, which was defined as ${PCMI}_{surr}$.

Step 7: G_PCMI was defined by comparing the deviation of the distribution between the original PCMI (${PCMI}_{original}$) and ${PCMI}_{surr}$. Statistical analysis involved the Wilcoxon signed-rank test. If $H_{0}$ = 1 and *p* < 0.001, G_PCMI = ${PCMI}_{original}$. Otherwise, G_PCMI = 0.

Step 8: A 60 × 60 symmetric connectivity matrix of G_PCMI was constructed to display functional connectivity in the local brain regions and across brain regions. In the study, the whole brain was divided into four local regions: the frontal region (FP1 FP2 AF7 AF8 F7 F5 F3 F1 F2 F4 F6 F8), the central region (FC1 FC3 FC5 FT7 FC2 FC6 FT8 C1 C3 C5 C2 C4 C6), the parietal region (CP1 CP3 CP5 TP7 CP2 CP4 CP6 TP8 P1 P3 P2 P4), and the occipital region (PO3 PO7 O1 PO4 PO8 O2). Across brain regions contained the frontal–parietal (F–P) region, frontal–occipital (F–O) region, and central–occipital (C–O) region.

### 2.8. Statistical Analysis

The Wilcoxon rank-sum test is a non-parametric test that is commonly used to detect significant differences in data that are not normally distributed. Here, this was test was applied to analyze the differences in G_PCMI before and after tDCS. Differences were statistically significant if *p* < 0.05.

Spearman’s correlation coefficient was used to indicate the degree of correlation between G_PCMI and CRS-R scores; *p* value was set as 0.05.

## 3. Results

### 3.1. Patients’ Characteristics and CRS-R Scores

Specific characteristics of patients are given in Table 2. Both of the two groups contained six MCS and four UWS patients (five males and five females in the frontal group; seven males and three females in the parietal group). In the frontal group, the mean age was 42.60 ± 11.36 years, the mean course of disease was 13.15 ± 12.25 months, and the baseline of the CRS-R score was 7.90 ± 1.97. For the parietal group, the mean age was 41.40 ± 16.02 years, the mean course was 6.80 ± 6.10 months, and the baseline of the CRS-R score was 7.70 ± 1.35. The CRS-R score did not show any change immediately after tDCS. However, at the 9–12-month follow-up, six patients in the frontal tDCS group showed an increase in CRS-R score; one patient showed new signs of emerging consciousness. In the parietal tDCS group, seven patients showed improvement in the CRS-R score; two patients showed new signs of consciousness. Both the frontal and parietal groups obtained significant increases in CRS-R scores (9.80 ± 3.55, *p* = 0.012; 9.00 ± 2.31, *p* = 0.006). However, there was no significant difference between the two groups (*p* = 0.227).

### 3.2. EEG Results

#### 3.2.1. G_PCMI before and after Frontal tDCS

Functional connectivity in local and across brain regions were included in a G_PCMI matrix (Figure 3). Figure 3a,b represent G_PCMI values before and after frontal tDCS, respectively. Figure 3c,d represent G_PCMI values before and after sham stimulation. The changes induced by tDCS were analyzed by comparing the G_PCMI values. After one session of 20 min frontal tDCS, the functional connectivity within the frontal regions was significantly enhanced (*p* < 0.05, Figure 4A); furthermore, the G_PCMI values in the frontal–parietal (F–P) region and frontal–occipital (F–O) region were significantly increased (*p* < 0.005, *p* < 0.05, Figure 4B). However, there were no significant changes in functional connectivity in local and across brain regions following sham stimulation (*p* > 0.05, Figure 4A,B).

#### 3.2.2. G_PCMI before and after Parietal tDCS

Results derived from the parietal tDCS group are shown in Figure 5. According to statistical analysis, the G_PCMI within the parietal region showed significant changes (*p* < 0.05, Figure 6A) after parietal tDCS. Across brain regions, there was a significant decrease in G_PCMI in the F–P region (*p* < 0.005), F–O (*p* < 0.005) region, and C–O region (Figure 6B). In comparison, one session of 20 min sham stimulation in the parietal region did not induce significant changes in G_PCMI (*p* > 0.05).

#### 3.2.3. G_PCMI of Different Outcome Groups before and after tDCS

We assigned the patients who received frontal or parietal tDCS into different groups according to the 9–12-month follow-up outcomes: a good-outcome group for frontal tDCS (the frontal responders), a poor-outcome group for frontal tDCS (the frontal non-responders), a good-outcome group for parietal tDCS (the parietal responders), and a poor-outcome group for parietal tDCS (the parietal non-responders). Then, we compared the changes in G_PCMI values between the frontal responders and the frontal non-responders as well as those between the parietal responders and the parietal non-responders. The frontal responders demonstrated a significant increase in three regions: the F–F, F–P, and F–O regions (*p* < 0.05, *p* < 0.05, and *p* < 0.005, respectively, as shown in Figure 7A). The frontal non-responders showed an increasing trend, but did not reach statistical significance (*p* > 0.05, Figure 7A). For the parietal tDCS, the parietal responders showed a significant difference in the F–P and F–O regions (*p* < 0.005 and *p* < 0.005, respectively, as shown in Figure 7B), while the parietal non-responders showed no significant changes (*p* > 0.05, Figure 7B).

### 3.3. Correlations between G_PCMI and CRS-R Values

#### 3.3.1. Correlations between the Changes in G_PCMI and Baseline CRS-R Values

For the frontal group, the changes in G_PCMI values in the F–F and F–P regions before and after tDCS were significantly correlated to the baseline CRS-R values (r = 0.67, 0.67; *p* < 0.05, Table 3). Significant correlations were evident in the P–P and F–P regions in the parietal tDCS group (r = 0.70, 0.66; *p* < 0.05, Table 3).

#### 3.3.2. Correlations between the Changes in G_PCMI Values and the Changes in CRS-R Values

Next, we investigated the correlations between the changes in G_PCMI values and the changes in CRS-R values (Table 4). We found that the changes in CRS-R values were significantly correlated to the changes in G_PCMI values in the F–F and F–P regions in the frontal tDCS group (r = 0.68, 0.75, *p* < 0.05). In addition, the changes in G_PCMI values induced by parietal tDCS were significantly correlated to the changes in CRS-R in the F–P and F–O regions (r = 0.72, 0.64, *p* < 0.05).

## 4. Discussion

To date, there is no effective way to arouse consciousness. Among various interventions, tDCS is a promising method. Compared with invasive neuromodulations (deep brain stimulation and spinal cord stimulation), tDCS has several advantages: there is no surgical risk, no ethical implications, and it is portable. In the tDCS protocol, the choice of target is always a key consideration. Despite the appearance of a number of theories over recent years regarding consciousness [33], the mechanisms underlying consciousness remain unknown. Of the four prominent theories, the Global Workspace Theory and the Higher-order Theory emphasize the importance of the frontal cortex in the generation of consciousness, while the Integrated Information Theory and the Recurrent Processing theory stress the importance of the posterior cortex [34]. However, there is no consensus regarding which cortex contributes more to consciousness. Studies involving non-invasive neuromodulation always consider the frontal cortex as the first target choice. Some researchers have shown that the posterior cortex may contribute more than the frontal cortex in terms of decoding differences in consciousness [34]. Thus, some studies have investigated the effect of parietal tDCS and achieved positive results. However, differences between these two tDCS targets in terms of neuromodulatory effects and underlying mechanisms remain unknown. In this study, we designed a scheme to compare the neuromodulation effects of frontal tDCS and parietal tDCS in the hope that we can provide a basis for clarifying the mechanism underlying these two targets. Meanwhile, we applied an EEG functional connectivity index, G_PCMI, to evaluate the effects of tDCS.

### 4.1. The Modulatory Effects of Frontal tDCS and Parietal tDCS

G_PCMI is a method used for non-linear analysis and can be used to characterize cortical functional connectivity [35]. Some studies have proven that the mutual information measure could distinguish different levels of consciousness, such as UWS, MCS, and consciousness state [36,37,38]. This is an effective index for evaluating the level of consciousness. In this study, we found that the G_PCMI significantly increased within the frontal regions, the F–P, and the F–O regions. However, in patients treated with parietal tDCS, the G_PCMI significantly decreased within the parietal regions and the F–P, F–O, and C–O regions. Sham stimulation showed no significant changes in G_PCMI values. In other words, analysis showed that functional connectivity increased in local and long-distance regions (the F–P and F–O regions) after frontal tDCS. This demonstrated that the frontal cortex could promote information interaction among cortices and help the brain regain functional activities. Parietal tDCS regulated brain function in a manner that was opposite to frontal tDCS in that it reduced G_PCMI values in the local parietal region and across brain regions—the F–P, F–O, and C–O regions. These opposing results between frontal and parietal tDCS mean that both frontal and parietal tDCS have modulatory effects in DoC; however, the mechanisms underlying these effects may differ.

### 4.2. The Prognosis of DoC Based on Responses to Frontal and Parietal tDCS

We classified the frontal and parietal tDCS patients as responders and non-responders according to the CRS-R scores at the 9–12-month follow-up. The frontal and parietal tDCS responders showed significant changes in G_PCMI, while the non-responders in both groups showed no significant changes. The long-distance connectivity after frontal tDCS was significantly related to the prognosis of consciousness state. This not only reflected the modulatory effects of tDCS, but also the connectivity and plasticity of the cortices. The generation of consciousness needs information mutuality; this is promoted by connectivity between the cortices. Plasticity was evident in the response of the cortices to stimulation. Similar to the outcomes of transcranial magnetic stimulation with electroencephalography (TMS-EEG), patients with a relatively preserved complete brain structure could achieve a better prognosis in that TMS could induce a response in the global cortex in patients with MCS, but could also induce local effects in patients with UWS patients. The enhancement of connectivity between distant regions reflects the preserved information mutuality and could provide a structural basis for the recovery of consciousness. Although the G_PCMI values decreased after parietal tDCS, the changes in G_PCMI values were related to the CRS-R values. This finding indicated that parietal tDCS also had neuromodulatory effects in DoC, although the mechanism underlying this effect was different to that of frontal tDCS. Thus, changes in G_PCMI could be used as a prognostic index to select the responders who might achieve improvements in consciousness over time.

Some studies [12,14,15,17] showed that frontal tDCS can induce an increase in connectivity among cortices in the theta, alpha, and beta bands and a decrease in connectivity among cortices in the gamma band. Some studies [27,29] indicated that parietal tDCS can induce an increase in connectivity among cortices in the beta and gamma bands and a decrease in connectivity among cortices in the delta band. However, many EEG indices can be used to characterize the connectivity among different cortices. Different studies have applied different indices to evaluate connectivity and investigate the associated mechanisms. To date, these studies have generated different outcomes, thus preventing a consensus of opinion. This study is the first to investigate the mechanisms responsible for the effects of frontal and parietal tDCS and verify the modulatory effects by applying the unified electroencephalographic index G_PCMI. Our analyses demonstrate that frontal tDCS and parietal tDCS can modulate the brain network to improve the levels of consciousness, although the mechanisms involved might be different between them. Frontal tDCS increased the values of G_PCMI, while parietal tDCS reduced the values of G_PCMI. Furthermore, we investigated the relationship between G_PCMI and CRS-R scores and found that the G_PCMI values and CRS-R scores are correlated. This indicated that G_PCMI represents an index to evaluate the efficacy of tDCS. Furthermore, we showed that the responders to tDCS exhibited significant changes in G_PCMI values after stimulation. Thus, G_PCMI could be used as a prognostic index to select the responders who might achieve improvements in consciousness over time. This will help clinicians optimize clinical decision making and save valuable medical resources.

### 4.3. Limitations

Although frontal tDCS and parietal tDCS achieved notable modulatory effects, there are still some limitations in this study that need to be considered. First, the number of sessions may represent one key restriction. In previous studies, a single session of tDCS therapy was not necessarily sufficient for the recovery of consciousness. In the present study, though CRS-R scores did not improve immediately after one session of tDCS, all patients showed improvement in CRS-R to various degrees at the 9–12-month follow-up. However, previous studies have confirmed the enhanced effects of multiple sessions of tDCS. Thus, a number of sessions need to be considered in future research. Second, the G_PCMI values changed significantly in the two tDCS groups; these changes were correlated to CRS-R scores. However, these changes occurred in two directions (frontal tDCS showed enhancement, while parietal tDCS showed the opposite effects). This might indicate that the mechanisms underlying these two targets are different. This study represented a preliminary investigation of the comparative effects of frontal tDCS and parietal tDCS; other comprehensive and diverse tools are needed to analyze the effects and clarify the mechanisms involved, such as neuro-imaging. Third, the small sample size and heterogeneity of patients are also potential limitations. Future research should involve a larger sample size from multiple centers. Finally, we did not take into account the natural recovery of patients. It would be a better design to validate the experiment if there was a control group (patients without tDCS).

## 5. Conclusions

A single session of frontal and parietal tDCS was applied to patients with DoC to compare the difference. Neither of them induced transient effects, but showed long-term effects at the 9–12 follow-up. However, the neurobehavioral effects between the two groups were no different. The G_PCMI is an EEG index than can be used to evaluate the neuromodulation effects of tDCS. Both frontal and parietal tDCS could induce significant changes in G_PCMI, but the underlying mechanisms might be different.

## Figures and Tables

**Figure 1 brainsci-13-01295-f001:**
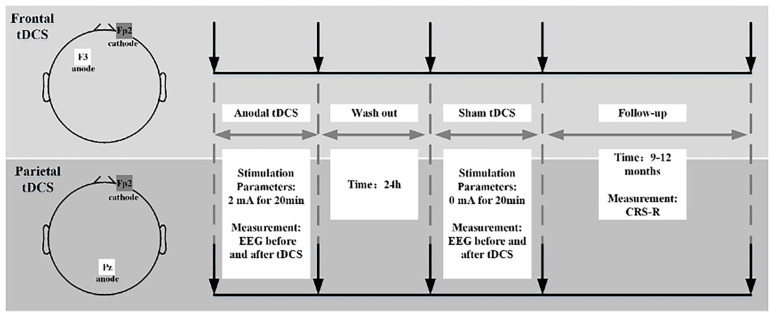
Flow chart showing the experimental design.

**Figure 2 brainsci-13-01295-f002:**
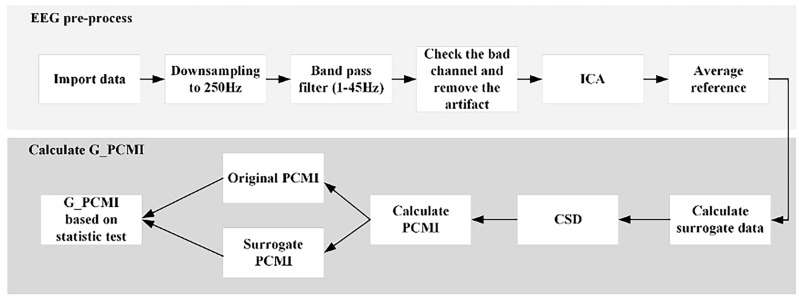
The process used to generate data and G_PCMI.

**Figure 3 brainsci-13-01295-f003:**
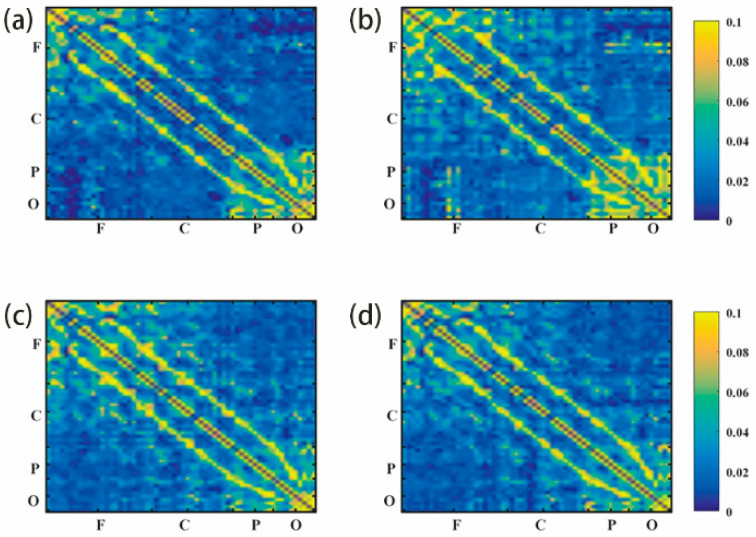
G_PCMI matrix before and after frontal tDCS: (**a**) G_PCMI matrix before frontal tDCS. (**b**) G_PCMI matrix after frontal tDCS. (**c**) G_PCMI matrix before sham stimulation. (**d**) G_PCMI matrix after sham stimulation. Different colors represent different values of G_PCMI. The brighter the color, the higher the value. The changes in color could reflect the changes in the G_PCMI. F, frontal region; C, central region; P, parietal region; O, occipital region.

**Figure 4 brainsci-13-01295-f004:**
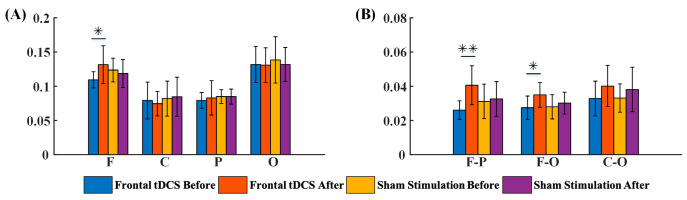
The G_PCMI of local and across brain regions before and after frontal tDCS, including the outcome of G_PCMI values by groups. (**A**) G_PCMI values in the local region before and after stimulation. (**B**) G_PCMI values across brain regions before and after stimulation. F, frontal region; C, central region; P, parietal region; O, occipital region; F–P, frontal–parietal region; F–O, frontal–occipital region; C–O, central–occipital region. *, *p* < 0.05; **, *p* < 0.005.

**Figure 5 brainsci-13-01295-f005:**
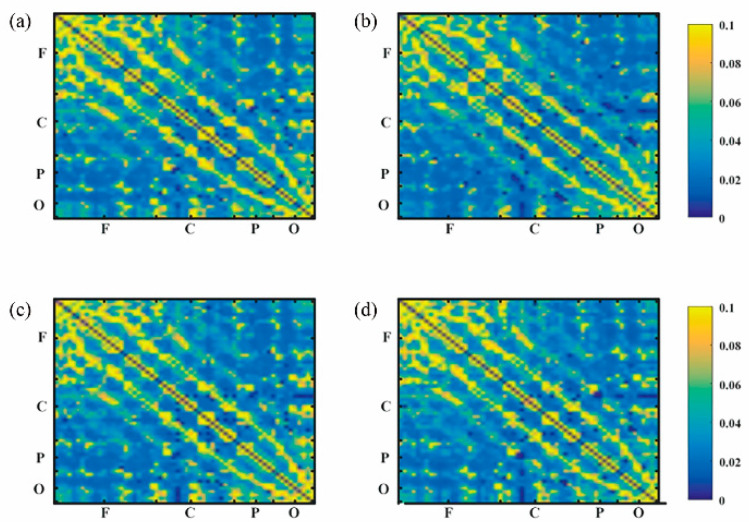
G_PCMI matrix before and after parietal tDCS: (**a**) G_PCMI matrix before parietal tDCS. (**b**) G_PCMI matrix after parietal tDCS. (**c**) G_PCMI matrix before sham stimulation. (**d**) G_PCMI matrix after sham stimulation. Different colors represent different values of G_PCMI. The brighter the color, the higher the value. The changes in color could reflect the changes in the G_PCMI. F, frontal region; C, central region; P, parietal region; O, occipital region.

**Figure 6 brainsci-13-01295-f006:**
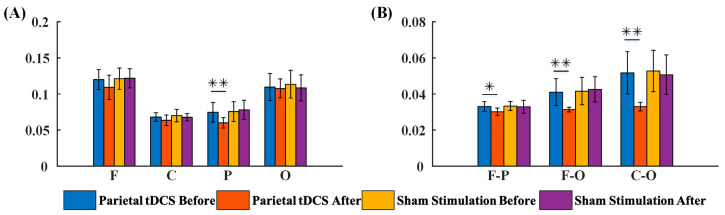
The G_PCMI of local and across brain regions before and after parietal tDCS, including the outcome of G_PCMI values by groups. (**A**) G_PCMI values in the local region before and after stimulation. (**B**) G_PCMI values across brain regions before and after stimulation. F, frontal region; C, central region; P, parietal region; O, occipital region; F–P, frontal–parietal region; F–O, frontal–occipital region; C–O, central–occipital region. *, *p* < 0.05; **, *p* < 0.005.

**Figure 7 brainsci-13-01295-f007:**
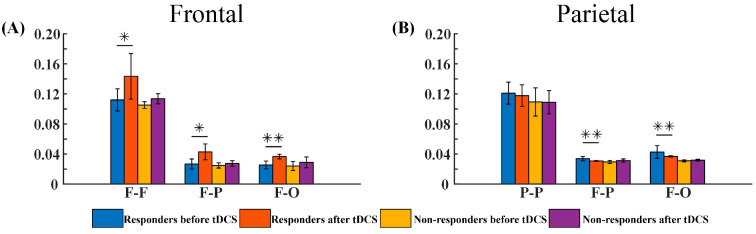
The G_PCMI values of different outcome groups before and after tDCS, including the outcome of G_PCMI values by groups. (**A**) G_PCMI values of the frontal responders and non-responders before and after stimulation. (**B**) G_PCMI values of parietal responders and non-responders before and after stimulation. F–F, frontal–frontal region; P–P, parietal–parietal region; F–P, frontal–parietal region; F–O, frontal–occipital region. *, *p* < 0.05; **, *p* < 0.005.

**Table 1 brainsci-13-01295-t001:** The general data relating to the patients with DoC in the frontal and parietal tDCS groups.

	Patients	Age (Year)	Gender	Etiology	Course (Month)	CRS-R (Baseline)	CRS-R (After tDCS)	CRS-R (Follow-Up)	Diagnosis (Follow-Up)
1	MCS	52	Female	Anoxic	36	8	8	8	-
2	MCS	31	Female	Traumatic	6	11	11	16	improved
3	MCS	47	Male	Hemorrhagic	2	9	9	9	-
4	UWS	52	Male	Hemorrhagic	30	6	6	6	-
5	UWS	50	Female	Anoxic	8	5	5	5	-
6	MCS	53	Male	Hemorrhage	8	11	11	15	MCS+
7	MCS	43	Male	Traumatic	8	9	9	11	improved
8	MCS	29	Female	Anoxic	28	7	7	9	improved
9	UWS	50	Female	Traumatic	4	7	7	11	improved
10	UWS	19	Male	Traumatic	1.5	6	6	8	improved
1	UWS	25	Male	Traumatic	7	6	6	8	MCS
2	MCS	24	Female	Traumatic	3	9	9	10	improved
3	MCS	60	Female	Anoxic	3	8	8	8	-
4	UWS	44	Male	Anoxic	3	7	7	8	improved
5	UWS	20	Male	Traumatic	3	6	6	6	-
6	UWS	42	Male	Ischemic	8	6	6	6	-
7	MCS	62	Male	Anoxic	3	10	10	13	improved
8	MCS	24	Male	Ischemic	24	8	8	9	improved
9	MCS	56	Male	Ischemic	8	9	9	12	improved
10	MCS	57	Female	Traumatic	6	8	8	10	MCS+

Abbreviations: DoC, disorders of consciousness; tDCS, transcranial direct current stimulation; CRS-R, coma recovery scale-revised; MCS, minimally conscious state; UWS, unresponsive wakefulness syndrome.

**Table 2 brainsci-13-01295-t002:** Patients’ characteristics.

Stimulation Target	Number	Age (Years)	Gender	Course of Disease (Months)	CRS-R (Baseline)	CRS-R (Follow-Up)
Frontal	10 (6 MCS, 4 UWS)	42.6 ± 11.36	5 males, 5 females	13.15 ± 12.25	7.90 ± 1.97	9.80 ± 3.55
Parietal	10 (6 MCS, 4 UWS)	41.4 ± 16.02	7 males, 3 females	6.80 ± 6.10	7.70 ± 1.35	9.00 ± 2.31

Abbreviations: tDCS, transcranial direct current stimulation; CRS-R, coma recovery scale-revised; MCS, minimally conscious state; UWS, unresponsive wakefulness syndrome.

**Table 3 brainsci-13-01295-t003:** Correlations between the changes in G_PCMI and baseline CRS-R values.

Location	Spearman Correlation	F–F	P–P	F–P	F–O
Frontal tDCS	r	**0.67**	-	**0.67**	0.53
	*p*	**<0.05**	-	**<0.05**	>0.05
Parietal tDCS	r	-	**0.70**	**0.66**	0.61
	*p*	-	**<0.05**	**<0.05**	>0.05

Abbreviations: F–F, frontal–frontal region; P–P, parietal–parietal region; F–P, frontal–parietal region; F–O, frontal–occipital region. Bold represents *p* < 0.05.

**Table 4 brainsci-13-01295-t004:** Correlations between the changes in G_PCMI values and the changes in CRS-R values.

Location	Spearman Correlation	F–F	P–P	F–P	F–O
frontal	r	**0.68**	-	**0.75**	0.21
	*p*	**<0.05**	-	**<0.05**	>0.05
parietal	r	-	0.15	**0.72**	**0.64**
	*p*	-	>0.05	**<0.05**	**<0.05**

Abbreviations: F–F, frontal–frontal region; P–P, parietal–parietal region; F–P, frontal–parietal region; F–O, frontal–occipital region. Bold represents *p* < 0.05.

## Data Availability

The data used and/or analyzed during the current study are available from the corresponding author on reasonable request.

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
