# Peer review of "A Comparison of the Neuromodulation Effects of Frontal and Parietal Transcranial Direct Current Stimulation on Disorders of Consciousness"

_brainsci, 2023, doi:10.3390/brainsci13091295_

Round 1
Reviewer 1 Report
Comments and Suggestions for Authors
The manuscript is very well written. The following improvements are needed:
- The figure legends need to be improved for Figures 3 and 5. Add more details in the figure legends describing the EEG results.
- Why was EEG not measured for 9-12 months follow-up?
- Authors should provide a comparison of DCS data with subjects not undergoing any DCS, i.e. How much do CRS-R scores improve for subjects with DoC after 9-12 months without any exposure to DCS? Authors should discuss if the results in this study are significantly different from the CRS-R scores for subjects that do not experience any DCS during 9-12 months of recovery.
Reviewer 2 Report
Comments and Suggestions for Authors
In a manuscript submitted for review, the Author described the comparison of the neuromodulation effects of frontal and parietal transcranial direct current stimulation on disorders of consciousness. I find the topic of the manuscript interesting and the whole work is thoughtful.
My comments:
1. there is no clearly defined purpose of the review of the experience, and above all, the conclusion does not answer whether the assumption from the purpose of the research was confirmed.
2. reference 1 and 34 are quite old. Is it necessary to include them in the manuscript?
3. did the Authors wonder what the results would be if men and women were compared separately?
Reviewer 3 Report
Comments and Suggestions for Authors
Comments to the Authors
Authors have investigated and compared the transcranial direct current stimulation in frontal and parietal brain areas clinically. The neuromodulation in both the groups (frontal and parietal) having disorders of consciousness were evaluated through CRS-R method in diverse age group of both genders. The study is well conduced. However, its significance needs more evidence. Here some comments.
1. Similar lines of articles related to transcranial direct cranial stimulation in psychiatric disorders including consciousness have already been published. The manuscript seems to lack much novelty, despite different brain areas being used.
2. The number of patients (20) are too less as sample to make a robust inference.
3. Could there be any control group without tDCS or a group with both prefrontal and parietal tDCS?
4. As the prefrontal cortex region of the brain is also involved in learning, memory, and decision-making, were these parameters considered during the study?
5. The grammar can be improved.
6. A few points are common to the limitation and Conclusion section. How about combining the two sections by briefing it?
7. Some the very old references can be updated, if possible.

English grammar must be improved.
